# Application of Super-Resolution and Advanced Quantitative Microscopy to the Spatio-Temporal Analysis of Influenza Virus Replication

**DOI:** 10.3390/v13020233

**Published:** 2021-02-02

**Authors:** Emma Touizer, Christian Sieben, Ricardo Henriques, Mark Marsh, Romain F. Laine

**Affiliations:** 1Division of Infection and Immunity, University College London, London WC1E 6AE, UK; emma.touizer.18@ucl.ac.uk; 2MRC Laboratory for Molecular Cell Biology, University College London, London WC1E 6BT, UK; r.henriques@ucl.ac.uk (R.H.); m.marsh@ucl.ac.uk (M.M.); 3Department of Cell Biology, Helmholtz Centre for Infection Research, 38124 Braunschweig, Germany; Christian.Sieben@helmholtz-hzi.de; 4The Francis Crick Institute, London NW1 1AT, UK; 5Instituto Gulbenkian de Ciência, 2780-156 Oeiras, Portugal

**Keywords:** super-resolution microscopy, advanced light microscopy, quantitative microscopy, live-cell microscopy, SMLM, STORM, SIM, STED, expansion microscopy, influenza virus, viral replication

## Abstract

With an estimated three to five million human cases annually and the potential to infect domestic and wild animal populations, influenza viruses are one of the greatest health and economic burdens to our society, and pose an ongoing threat of large-scale pandemics. Despite our knowledge of many important aspects of influenza virus biology, there is still much to learn about how influenza viruses replicate in infected cells, for instance, how they use entry receptors or exploit host cell trafficking pathways. These gaps in our knowledge are due, in part, to the difficulty of directly observing viruses in living cells. In recent years, advances in light microscopy, including super-resolution microscopy and single-molecule imaging, have enabled many viral replication steps to be visualised dynamically in living cells. In particular, the ability to track single virions and their components, in real time, now allows specific pathways to be interrogated, providing new insights to various aspects of the virus-host cell interaction. In this review, we discuss how state-of-the-art imaging technologies, notably quantitative live-cell and super-resolution microscopy, are providing new nanoscale and molecular insights into influenza virus replication and revealing new opportunities for developing antiviral strategies.

## 1. Introduction

In humans, influenza viruses cause acute infections in the respiratory tract. Every year, three to five million people develop severe seasonal influenza with up to 650,000 deaths, globally [1]. Although vaccines and some antivirals (e.g. Tamiflu) against seasonal influenza are available, the rapid antigenic drift and shift have [2], to date, made it difficult to obtain broadly neutralizing vaccines effective against multiple viral strains. Due to the alarming pandemic potential of emerging zoonotic influenza viruses [3], the WHO recently listed a ‘Global Influenza Pandemic’ as one of the top 10 threats to global health [4]. The emergence of SARS-CoV-2 and its associated pandemic [5] has greatly emphasised the need to better understand highly transmissible respiratory viruses, including influenza virus.

Influenza virions consist of an RNA genome encased in a lipoprotein membrane (the viral envelope). The viral genome is made up of eight RNA segments, wrapped by the viral nucleoproteins and the RNA polymerase (RNApol made of PB1, PB2 and PA), termed viral ribonucleoprotein complexes (vRNP). The envelope is derived from the host cell plasma membrane and contains two major glycoproteins, hemagglutinin (HA) and neuraminidase (NA), together with the proton channel M2. The M1 protein forms a proteinaceous matrix underneath the envelope, as seen in Figure 1a. HA and NA are essential for infection as they mediate the first steps of entry. HA binds directly to receptors decorated with sialic acid to promote viral entry, whereas NA is capable of cleaving sialic acid from glycoproteins, and is thus mainly involved in viral release from the plasma membrane upon budding as well as penetration of mucus on the respiratory epithelium. As a consequence, the balance between the amount of HA and NA in the viral envelope has been shown to be critical for virus fitness [6]. From a therapeutic point of view, both HA and NA are exposed on the surface of virions making them key anti-viral and vaccine targets. However, antigenic shift and drift are the main drivers for influenza virus diversity and lead to the limited efficacy of annual influenza vaccines [7].

To understand the mechanisms underlying influenza virus replication and of cell-to-cell viral transmission, with a view to developing new anti-viral strategies, one emerging requirement is to be able to directly observe virions and virus-host cell interactions. The small size of influenza virions (~120 nm diameter) has made electron microscopy (EM) the method of choice for these observations. For example, EM has revealed that influenza virus particles can be pleiomorphic, taking shapes from ~120 nm diameter spherical particles to long filamentous forms up to several µm long and ~90 nm in diameter (reviewed in [8,9]), but the relevance of this pleiomorphism to viral infection and pathogenesis in vivo is currently unclear. Cryo-EM has been instrumental in structural studies, showing, for example, that M1 forms a helical array that supports the shape of filamentous particles [10], revealing details of the structure of HA molecules [11], and that the organisation of HA is influenced by the curvature of the viral membrane. This highlights that the lateral distribution and density of HA on the virus are key for entry [12]. Although EM can resolve viral and cellular structures at near-atomic level, its use is usually limited to fixed samples, only providing a snapshot in time of specific events and no, or only low levels of, molecular identification. In contrast, while typically limited to ~250 nm resolution due to the diffraction of light, optical imaging combined with fluorescence has high molecular specificity and is compatible with live-cell imaging. 

In recent years, the development of advanced and quantitative fluorescence microscopy and, importantly, that of super-resolution microscopy (SRM) [13] has opened new possibilities for direct imaging of cells and for understanding host-pathogen interactions by providing a powerful combination of enhanced spatial resolution, very high molecular specificity and practical compatibility with live-cell imaging [14]. As an example of the potential of SRM to reveal new insights to viral biology, its application to studies of human immunodeficiency virus (HIV) (reviewed in [15,16]) has revealed new insights into the nanoscale distribution of the viral envelope protein (Env) on the surface of virions [17] and the role of ESCRT complexes in HIV egress [18].

In this review, we present examples of the exciting new advances in the study of influenza virus permitted by novel optical imaging tools. We limit our scope to SRM, single-molecule imaging and quantitative live-cell analysis, with a particular emphasis on SRM, since observing virus assemblies at the nanoscale is central to understanding virus replication. Therefore, we first describe these technologies and then present recent studies in the context of the different aspects of influenza viral replication, from entry to viral budding. We finally provide an outlook of what these techniques may unravel in the future.

We identified three key opportunities for advanced fluorescence microscopy techniques to provide novel insights into influenza virus biology, as highlighted in Figure 1b–d: (i) Understanding virus structures and associated host cell components at the nanoscale [17,18,19,20], (ii) resolving individual viruses among large populations of viruses [21,22] and (iii) observing dynamic processes in real-time [23,24].

In Figure 1b, the use of correlative EM and three-dimensional (3D)-SRM allowed the distribution of HA in filamentous influenza viruses to be observed at the nanoscale [19], corroborating the structural information from EM to the chemical identity information at the single-molecule level from the SRM technique used here. The nanoscale resolution achieved by SRM is capable of giving information at the molecular organisation level, highlighting specific nanoscale assemblies both in the viral structure [17,25] and in the infected cell [26]. Observing large virus populations at the single-virus level can also be used to directly investigate the diversity of structures within pleiomorphic virus population [21] or the functional effects of drugs, as seen in Figure 1c. Adding the NA inhibitor oseltamivir reduces the release of filamentous viral particles, thus, highlighting a virus shape-specific effect of the drug [22]. Live-cell imaging can offer a wide range of possibilities to observe the replication cycle of influenza virus. In Figure 1d, fluorescence light-sheet microscopy in living cells combined with quantitative particle tracking in 3D showed the co-transport of Rab11A-containing recycling endosomes with influenza viral RNA (vRNA) and the exploitation of this trafficking pathway by the virus [24].

## 2. Overview of Super-Resolution Microscopy and Novel Imaging Methods to Study Influenza Virus Replication

SRM comprises optical imaging techniques that offer resolution beyond the diffraction limit of light (~250 nm). Figure 2 shows graphical explanations and examples of the main SRM techniques applied to the study of influenza virus replication.

SRM was developed in the mid-1990s, first with stimulated emission depletion (STED) microscopy [27]. STED is a scanning technique based on laser confocal scanning microscopy, which exploits the phenomenon of stimulated emission to quench the fluorescence signal around the excitation scanning point. This effectively reduces the volume of the fluorescence measurement and typically improves resolution down to 50–70 nm. For influenza virus, the resolution gain of STED has been used to follow, for example, vRNPs during viral trafficking along the endosomal pathway in dendritic cells (Figure 2a; [28]). By allowing nanoscale localization of structures, STED provides a powerful way to determine the identity and the structure of the organelles involved in vRNP trafficking with high spatial precision.

A second technique called structured illumination microscopy (SIM) [29] relies on the acquisition of a sequence of fluorescence images obtained under a set of structured illuminations (typically made of stripes), each creating Moiré patterns with the underlying structure of interest. The measured Moiré patterns contain information about high spatial frequencies (high-resolution information) in the sample that can be recovered by image reconstruction of the raw data, typically achieving resolution down to 150 nm. In Figure 2b, 3D SIM imaging is used to show that ubiquitin is packaged inside purified influenza virions [30]. This study showed that ubiquitin is then used during virus entry to recruit the cellular aggresome pathway to facilitate capsid disassembly and release of vRNPs. Despite the limited resolution increase of SIM, it has been successfully used to describe the substructure of viruses, such as vaccinia [31] and the heterogeneous morphology of influenza virions [21]. 

A third approach to SRM is to exploit the capability of certain fluorescent dyes to blink (photoswitch) under specific experimental conditions. This set of techniques, collectively called single-molecule localization microscopy (SMLM), relies on the use of fluorophores that can be effectively switched on and off in a stochastic manner. By isolating single fluorophores spatio-temporally, it is possible to precisely localize them in space despite each molecule producing a diffraction-limited spot on the camera, and to then build a nanoscale map of fluorophore distribution. SMLM techniques routinely achieve a localization precision of 10–20 nm (Figure 2c). Some of earliest described SMLM techniques, PALM (photo-activated localization microscopy) [32] and fPALM (fluorescence PALM) [33] use genetically engineered photo-activatable fluorescent proteins, whereas STORM (stochastic optical reconstruction microscopy) [34] uses conventional photo-switchable organic dyes such as Cy5. Variations on these original SMLM approaches have been developed including *d*STORM (*direct* STORM) [13,35], PAINT (points accumulation for imaging in nanoscale topography) [36] and GSD (ground state depletion) [37]. 

An exciting variant of these latter techniques is single-particle tracking PALM (sptPALM), a technology capable of deciphering the spatial organisation and dynamics of individual molecules by randomly photo-activating single-molecules and tracking them in living cells. This approach was originally demonstrated by tracking individual HIV Gag proteins at the plasma membrane of Gag expressing cells [38].

During influenza virus assembly, it was shown using a biochemistry approach that HA preferentially accumulates in so-called lipid rafts [41,42], mediated through its transmembrane domain [43]. Some of the earliest fPALM experiments, which looked at this association at the plasma membrane of HA-expressing fibroblasts [44], could visualize that HA forms irregular, lipid raft-associated clusters with a similar size range to that of budding virions. The high-resolution of live-cell fPALM enabled different models of membrane organisation to be discerned and revealed the molecular dynamics within the clusters. Further, an early form of SMLM called ‘Blink’ [45] showed that HA at the plasma membrane of infected cells forms dynamic nanodomains of around 80 nm [46]. The small size of these microdomains would be challenging to observe with conventional fluorescence imaging techniques.

Figure 2c shows a filamentous influenza virus imaged with *d*STORM. The nanoscale resolution achieved by *d*STORM reveals the spatially alternating distribution of HA and NA-rich regions along a filamentous influenza virion.

A newcomer in the SRM field is expansion microscopy (ExM) [47]. This method turns the diffraction limit problem on its head by expanding the sample isotropically in a hydrogel, practically improving the resolution of images by the expansion factor, typically of ~4× [47] and up to ~20× with iterative ExM (iExM) [48]. The approach can even preserve and resolve the integrity of bio-macromolecular assemblies, as successfully demonstrated by visualizing the molecular organisation of the centrosome [49]. In Figure 2d, we show an example of ExM used to study the spatial organisation of cytoskeletal structures in cells. Although some early studies show that ExM can be used to study viral infection and viral assembly [50,51,52], including for bacteriophage T5 [53], ExM remains under-used in the context of virology despite its potential versatility and ease of use.

Although each of these techniques provides nanoscale imaging, they also have their own advantages and drawbacks which need to be considered when choosing a method. If maximum resolution is required and the imaging is performed on fixed cells, then SMLM can provide an optimal solution. On the other hand, when imaging tissue or thick samples, STED can be an effective method due to its inherent optical sectioning capabilities. SIM, on the other hand, provides an easy and fast solution for live-cell imaging at medium resolution. ExM is not compatible with live sample imaging but constitutes a sample-based approach to SRM that can easily be combined with the other SRM techniques, and thus, provides a novel way to enhance resolution in tissue in 3D.

Beyond SRM, quantitative long-term live-cell imaging has great promise for the study of host-pathogen interactions at relevant temporal and spatial scales. For instance, by being able to visualize viruses and their components interacting with host cells, quantitative live-cell fluorescence microscopy can be used to study the cellular pathways exploited by viruses at the single-cell and single-virion level [8,9,22]. SRM and quantitative live-cell fluorescence microscopy was used to study the uncoating of HIV in living cells with a high temporal resolution providing evidence that the HIV capsid can remain intact while entering the cell nucleus [23]. The study of live-cell dynamics has been further enhanced through light-sheet microscopy, which allows high speed, long-term imaging in 3D with low phototoxicity [54], but this method remains largely underexploited in the context of viral replication, potentially due to the lack of availability of such tools in appropriate containment conditions for live virus imaging. 

Further, recent advances in microscopy sensitivity for the detection of single molecules, and the introduction of new and non-invasive labelling strategies such as FlAsH [55], Sfp [56], transglutaminase 2 [57], sortase A [58] with bright fluorescent markers, offer new opportunities for improved live-cell imaging. These new technologies enable the behaviours of viral and cellular components to be mapped dynamically in super-resolution [14], multi-label structural studies using intact viruses [22] and single-particle tracking (SPT) to follow individual virus particles during infection and individual viral components during replication [59,60,61]. In the context of SPT, a powerful approach has been to use quantum dots (QDs) [62,63,64], for long-term imaging, but with limitations due to the relative complexity of labelling steps. 

Single-molecule Förster Resonance Energy Transfer (smFRET) [65] constitutes a powerful optical method to observe nanoscale changes in conformations within biomolecules, especially when using small peptide labels and quantitative analysis. In the context of influenza, this approach has been used to study the dynamics of fusion-associated low pH-induced HA conformational changes [66]. The direct spatial localization of individual RNA transcripts can be performed by single-molecule fluorescence using in situ hybridization (FISH) [67]. FISH is also well-suited to studies of the assembly of influenza virions and has been used to observe vRNPs in the cytoplasm en route to budding sites [68,69].

We note that these imaging approaches can also benefit from quantitative approaches such as single-particle averaging [21,25,31], machine learning [70,71] and modelling [21,25,72], in order to tease out the most informative features from the imaging data.

## 3. Understanding Virus Structure 

To study the structure and composition of influenza viruses, it can be useful to combine nanoscale resolution and chemical specificity, SRM is perfectly suited to this task. Since such studies are typically carried out using purified viruses, SMLM and STED have exceptional potential. For instance, correlative STORM and EM have confirmed that SRM can efficiently describe the nanoscale distribution of HA, M1 and vRNP in filamentous influenza virions [19]. Here, the technique provided sufficient resolution and specificity to distinguish HA inserted into the viral envelope on each side of the ~100 nm diameter tubular core structure and confirmed previous observations that the vRNPs are located at the distal end of budding filaments (Figure 1a), in agreement with previous EM observations [10]. The wider availability of SRM microscopes compared to EM, together with the multi-colour and high-throughput capabilities underpin the potential of SRM as an essential tool for structural studies for viruses.

Multi-colour fluorescence microscopy makes it possible to spatially resolve many different components of a virion. However, in order to effectively image these structures, there is a need for efficient labelling of the biomolecules of interest. Although immunofluorescence provides a robust way to label fixed samples, live virus imaging can prove challenging. For instance, labelling viral components with large genetic tags, such as GFP, can compromise the infectivity of the engineered viruses. In such cases, novel small-peptide labelling strategies can provide a great solution. A recent study used innovative site-specific labelling strategies, combining enzymatic approaches and small size tags, that were minimally disruptive, to label 8 different influenza virion proteins (HA, NA, NP, N, M, PA, PB1 and PB2) to study both the morphology and the replicative fitness of live viruses (Figure 3a) [22]. Interestingly, the authors observed that a cell infected by a single virus particle could produce progeny virions with considerable heterogeneity in shape and composition. This study also showed that elongated viruses are polarized with one NA-rich pole and one HA-rich pole. This polarity was found to be crucial for influenza virus motility on pathogenically relevant mucosal surfaces. Further, *d*STORM imaging allowed the observation of the nanoscale distribution of HA and NA on filamentous viruses [39], showing small clusters of NA along the length of the filament in an alternating pattern with HA (see Figure 1c). The biological relevance of this precise pattern is currently unclear, but the authors provided evidence through modelling that specific spatial HA and NA organisation can support the clearing of mucosal surfaces by the virus through the unique combination of binding and cleaving of sialic acid groups. 

SRM microscopy techniques can also be combined with advanced quantitative analysis to quantify the structural diversity of virions within a virus population. In particular, TIRF-SIM (total internal reflection fluorescence SIM) was used in combination with machine learning to develop a high-throughput methodology capable of classifying influenza viruses based on their shape [21]. This automated framework has been used to study different strains of live-attenuated influenza viruses (LAIV) to understand potential associations of form with functionality.

Although STED has not been used to study the structure of individual influenza viruses, it has allowed the demonstration that HIV envelope protein (Env) distribution on virions is regulated by the structural polyprotein Gag proteolysis/maturation, a rearrangement that is key for efficient receptor engagement and viral entry [17]. Additionally, STED has been used in combination with fluorescence correlation spectroscopy (FCS) [73] to study the diffusion properties of Env proteins during this maturation process within individual HIV particles [74]. This study of dynamic processes within a virus highlights the potential of SRM for the study of structural re-organisation during infection, especially prominent during virus uncoating. Together, these studies illustrate the potential of STED microscopy for the structural analysis of viruses.

## 4. Understanding Viral Entry and Trafficking

The entry of the influenza viruses into cells is a multi-step process. Following the engagement of HA with sialic acid-decorated cell surface receptors, influenza virions undergo endocytosis through clathrin-mediated mechanisms [77], or macropinocytosis [78], with some possible cell dependence [79]. Although HA will bind various sialylated glycoproteins and glycolipids, the engagement of specific cell surface glycoproteins appears to be necessary for successful infection. The identity of these receptors is still a source of debate [80,81], but candidate proteins include the epidermal growth factor receptor (EGFR) that triggers clathrin-mediated endocytosis upon activation by influenza virus [82]. Following uptake from the plasma membrane, influenza viruses traffic through early endosomes to late endosomes where the acidic pH activates (1) the viral membrane proton channel M2, which triggers conformational changes in the viral core required for efficient uncoating [83,84]; and (2) HA-mediated fusion with the endosomal membrane [83]. Importantly, late endosomes are located close to the nuclear periphery, thus, uncoating is spatially coupled to the import of vRNPs into the nucleus for replication.

Advanced microscopy, and especially SPT, has provided essential insights into the dynamics and pathways exploited by influenza virus for entry [85]. Here, the lipophilic dye DiD, that inserts directly into the viral membrane, was used. Individual virions were tracked and quantified in living cells to demonstrate that similar percentages of influenza particles colocalized with Epsin-1, a cargo-specific adaptor for clathrin-mediated endocytosis, and clathrin, hinting that Epsin-1 may promote clathrin-mediated endocytosis. Recently, a similar live-cell SPT study suggested that influenza virus dynamics at the plasma membrane of target cells are determined by the spatial distribution of sialylated glycans, with single influenza particles dwelling for longer periods of time on glycan-rich regions [26]. This highlighted the existence of regions of the plasma membrane where the virus may have more time to engage with receptors and trigger endocytosis. In the same study, dual-colour STORM was used to measure the size and density of the sialylated glycan domains, as well as the nanoscale organisation of EGFR at the plasma membrane. This showed that EGFR forms nanoclusters that partially colocalize with the sialylated glycans domains (Figure 3b), an association that may provide enough time for EGFR clusters to initiate endocytosis at the virus binding site. This study shows that a combination of advanced microscopy techniques can lead to a complementary set of dynamic and structural information that builds a more complete picture of the biological processes.

Within early endosomes, influenza viruses exploit retrograde trafficking along the cell’s microtubule network to reach perinuclear late endosomes where fusion and uncoating occurs. Vesicular identity (e.g., early or late endosomes) can be obtained by simple colocalization microscopy studies, using well-established markers, and this can lead to a better understanding of the dynamics of trafficking. However, the higher resolution provided by SRM and colocalization at the nanoscale can discriminate between virions exploiting specific endosomal carriers as opposed to being in close proximity. In addition, SRM can be used to observe possible nanoscale structural re-organisations of endosomes and other organelles as a consequence of viral infection. Focusing on early trafficking events through the endosomal pathway, STED was exploited to study the entry dynamics of spherical influenza particles in human dendritic cells [28] (see Figure 2a and Figure 3c). However, this study did not explore structural changes of the endosomal system and was performed in fixed cells at specific time points post-infection, though it is technically possible to do this in living cells with SRM.

Recently, the use of QDs was shown to provide a powerful way to label viral components for long term tracking within living cells, allowing the dynamics of viral infection to be studied in near real-time over extended periods. This approach was used to label the viral envelope and track viruses inside living infected cells as they traffic along the microtubule network towards the perinuclear region, revealing the complex range of motion exhibited by the virus during trafficking towards the nucleus, and highlighting the impact of the virus on molecular transport processes within infected cells [62,63].

SRM can, thus, be used to interrogate multiple aspects of virus entry and trafficking including the identity and organisation of viral receptors, the identity of specific organelles and dynamics of the various events. However, the modulation of these phenomena with respect to dynamics and viral structure remain poorly understood despite viral entry being a critical step to target for anti-viral therapy.

## 5. Understanding Fusion and Uncoating

Upon reaching the perinuclear region of a cell, influenza viruses need to release their genetic material into the cytoplasm. This release requires the fusion of the viral membrane with the limiting endosomal membrane and disassembly of the M1 proteinaceous layer lining the inside of the viral envelope (uncoating). The viral envelope fusion with the endosome membrane is HA-mediated and occurs through a multi-step process involving a hemifusion intermediate, where only the apposed leaflets of the membranes merge, followed by the formation of a pore and finally complete fusion. 

To study the dynamics of fusion under the microscope, it is necessary to have access to a readout of the state of the viral envelope in real-time. A common approach involves observing the dequenching of the fluorescence signal from a lipophilic dye incorporated into the viral envelope [59]. For instance, an in vitro study on supported lipid bilayers used two-colour TIRF to show the kinetics of influenza virus membrane fusion at the single-particle level [86]. The interior of the virus particles was loaded with a fluorescent marker and the envelope labelled with a lipophilic dye, which remained quenched in intact particles. Dequenching of the lipophilic dye indicated the formation of hemifusion intermediates while the release of the dye from the interior of the virus indicated full fusion. By tracking changes in the intensity of both dyes over time, the authors were able to quantify the precise timing of different phases of the fusion reaction from hemifusion to pore-formation. A similar approach was also demonstrated in cells, though without the possibility to pinpoint hemifusion [87]. Although this approach does not constitute a recent imaging methodological development, it remains a powerful and commonly used microscopy tool that can benefit from analytical methods such as SPT and probe development for longer-term imaging.

By contrast, recent smFRET revealed the dynamics of fusion associated conformational changes of HA in intact virions [66]. The study used single-molecule sensitivity, small peptide labels and quantitative analysis to show that HA is capable of sampling a range of conformations in its pre-fusion state, but its equilibrium is shifted towards a more fusion-favourable state upon acidification. Such an approach can reveal excellent temporal resolution in vitro using TIRF microscopy but has not yet been demonstrated in a cellular context. Potential optical configurations such as light-sheet microscopy may offer the single-molecule sensitivity for such assays in infected cells [88].

Following fusion, influenza virus uncoats progressively: first the matrix layer of M1 proteins dissociates followed by the release of the 8 vRNPs into the cytosol. A recent study has shown that influenza viruses can exploit the cellular aggresome to promote uncoating following M2-mediated acidification of the viral core [30]. For this, 3D-SIM was used to show the presence of unanchored ubiquitin chains packaged into virions (see Figure 2b) that recruit histone deacetylase 6 (HDAC6) to fusion sites on late endosomes to help influenza uncoating. In a follow-up study, the authors demonstrated that the karyopherin TNPO1 is also involved in uncoating by promoting the dissociation of M1 and disassembly of the vRNPs [89]. Specifically, SIM was used to observe TNPO1-associated de-bundling of vRNPs in the cytoplasm before nuclear import. Here, the resolution improvement of SIM was sufficient to show the molecular associations involved in these steps of uncoating. Also, the compatibility of SIM with a wide range of labelling approaches makes SIM an excellent tool to complement biochemical studies.

After uncoating in late endosomes, vRNPs need to travel through the cytoplasm and into the nucleus where viral genome replication and expression occur. Mathematical modelling of the key steps of viral trafficking suggested that vRNP uptake into the nucleus was shown to be a critical bottleneck for efficient infection due to the potential for degradation of incoming RNA in the cytoplasm [72]. The model was calibrated using a combination of biochemical assays, cytometry and microscopy. Quantitative live-cell microscopy may allow the refining such modelling approaches to provide a fully quantitative biophysical framework for viral infection.

The diffusion of the vRNPs from the release site to the nucleus has been studied using SPT and a QD-labelling strategy [64] (Figure 3d). By using multi-colour QD labelling, this study was able to elegantly show that vRNP segments are released sequentially from late endosomes during uncoating and enter the nucleus through a multi-stage movement process. 

Thus, the capacity of advanced quantitative live-cell imaging to perform fast-tracking of multiple viral components simultaneously can provide dynamic information that enhances our understanding of the underlying biophysical processes. Fast, sensitive live-cell imaging of early infection events from trafficking, nuclear import to uncoating was also used in an elegant study of HIV uncoating [23]. Specifically, this study provided the first direct demonstration that the HIV capsid crosses the nuclear envelope intact, supporting the notion that capsid integrity is essential to protect HIV from innate immune sensing.

Although fusion and uncoating involve fundamental structural reorganisations of viruses at the nanoscale, SRM has yet to be used to observe these phenomena in the context of influenza. This may be a consequence of the difficulty of observing these events as they occur since the highest-resolution SRM techniques like SMLM mainly work in fixed rather than live samples. 

## 6. Understanding Assembly, Budding and Release

To produce new infectious virions, the essential viral proteins and the viral genome must be transported from their sites of replication, the cytoplasm, ER and nucleus, respectively, to the plasma membrane where virions assemble. To study the trafficking events leading to virus assembly, it is necessary to be able to follow the spatio-temporal dynamics of viral components. It is particularly important to understand how the segmented genome is capable of bundling into infectious virions, since genomic reassortment is linked to the appearance of emerging viral strains, with pandemic potentials.

In this regard, a seminal technical development has been multi-colour FISH [67] approaches to describe the composition and specific-bundling of the 8 unique vRNP segments in influenza virions [68,69,90]. Other methods are typically based on tomography [91,92,93] or molecular genetics approaches [94,95]. Multi-colour FISH enables the stoichiometry and the molecular interactions involved in influenza virus reassortment to be assessed in easier and more quantitative ways compared to other approaches. Recent advances in multiplexing FISH have allowed up to four segments to be studied simultaneously and shown that multiple segments are capable of forming subcomplexes throughout the cytoplasm during infection [96]. Another recent development called Multiple Sequential FISH (MuSeq-FISH), based on sequential single-molecule imaging of vRNA segments, has enabled a large-scale colocalization study between the 8 vRNA segments during influenza assembly [75] (see Figure 3e). By observing the relative occurrences and routes of specific bundling of vRNPs in infected cells, combined with mathematical modelling, a set of possible bundling orders of the vRNPs was suggested. Here, the authors highlighted that this selective packaging may be central to RNA re-assortment and its associated pandemic potential. Again, the combination of single-molecule imaging and advanced quantitative analysis can provide clear biophysical insights into the mechanisms of viral assembly. 

On the other hand, advanced live-cell imaging can provide direct information about the dynamics of transport of viral component to assembly sites and the mechanisms involved. Fast 3D tracking using light-sheet microscopy was exploited to follow the co-transport of multiple influenza vRNA segments (using GFP-tagged PA) [96] and the recycling endosomal marker Rab11A (using mRuby-tagged PA) [24] in living cells (see Figure 1c). Following vRNA transport in 3D at high speed enabled parallelisation of tracking observations captured in the cell volume. These studies confirmed that assembly of vRNPs can occur en route to the budding site, facilitated by the association with and modulation of Rab11A-associated vesicular transport, as previously shown [97,98], and indicating the profound effect of infection on molecular and vesicular transport dynamics [24].

In some cases, following the assembly process in living cells through the visualisation of newly expressed viral proteins can constitute a challenge for fluorescence labelling since genetic tagging of viral proteins can lead to loss of infectivity and/or assembly defects. Therefore strategies for non-invasive labelling (i.e., producing infectious viruses) of newly expressed viral proteins were designed. Notably, a split green fluorescent protein (GFP) approach [99] was used to fluorescently tag the influenza polymerase PB2 protein [100], in order to observe vRNPs trafficking in live infected cells. This labelling strategy was shown to be minimally disruptive and allowed the accumulation of nascent vRNPs at the plasma membrane, alongside PB2, to be observed in live cells.

Influenza virus assembly, budding and release is also closely tied to HA, NA and M2 distribution on the host cell plasma membrane [101]. Previous studies have shown that whilst HA and NA are found in lipid rafts [41], M2 is excluded [102], despite the fact that M2’s location at budding sites is key for membrane scission [103]. Therefore, the organisation and regulation of these microdomains, containing HA/NA/M2 is critical for all steps of viral assembly and release. Here, SRM can provide a powerful solution to studying the nanoscale organisation of viral components at and around the site of assembly. Live-cell fPALM combined with quantitative spatial correlation analysis showed that the cluster distribution of HA in the plasma membrane of infected cells correlates positively with dense regions of actin and negatively with cofilin at the nanoscale [104]. Single-molecule tracking approaches also showed that the mobility of HA diffusion was reduced in actin-rich regions. These findings point to a potential two-way feedback between HA clustering and actin dynamics. A further study using the same techniques recently revealed that nanoscale clusters of HA confine the diffusion of PIP2 [105], suggesting that HA acts on intracellular pathways involved in PIP2 regulation of actin re-modelling.

Interfering with these dynamic microdomains can potentially be used as a therapeutic intervention. STORM imaging of budding filamentous influenza viruses revealed that antibodies against the extracellular part of M2 (M2e) were able to disrupt virus production [106]. Interestingly, the study also observed the presence of HA-filled protrusions connecting infected cells with neighbouring uninfected cells, suggesting a potential role for filamentous viruses in direct cell-to-cell infection.

Advanced microscopy can be used to investigate the role of specific host proteins during infection. A study unveiled CD81, a tetraspanin previously shown to be a co-receptor for hepatitis C virus entry [107], as a key host factor for influenza uncoating and virus release [76]. This study showed that CD81 is recruited to influenza virus budding sites (Figure 3f) and is found clustered in nascent virions during budding. Using 3D STORM, the authors found that CD81 clusters in an alternating fashion with PB1 (every 150–200 nm) on the surface of budding filamentous viruses. Although the importance of this pattern, if any, is currently unknown, SRM helped reveal the nanoscale co-organisation of a host protein, CD81, with a viral component, PB1.

It is worth highlighting that SRM has proven very powerful in the study of budding mechanisms for HIV. In particular, PALM and *d*STORM have been used to study how Env is recruited to Gag protein clusters during HIV assembly at the plasma membrane [108], and 3D SMLM to delineate the distribution and potential role of the ESCRT machinery in and around HIV budding sites [18,20]. More recently, a similar 3D SMLM strategy allowed the nanoscale observation that HIV Env does not distribute randomly within the budding virion but rather accumulates around the neck of the nascent virions, suggesting that Gag lattices form prior to Env assembly [20]. This observation was also later supplemented by single-molecule tracking of Env at the budding site, showing that Env cytoplasmic tail is necessary for its interaction with Gag [109]. These studies show that advanced SPT and SRM approaches can help to identify mechanisms of infection and virus host-cell interactions at the molecular level. Their application to influenza virus may contribute to understanding the host-cell interactions that determine viral shape.

## 7. Understanding Viral Restriction by the Host Cell and Supporting the Development of Antiviral Strategies

Host cell restriction and innate immune responses are important intrinsic suppressors of viral replication. The interferon-induced antiviral protein IFITM3 has previously been identified as an influenza restriction factor that inhibits viral entry prior to uncoating [110] and increases viral degradation through the lysosomal pathway [111]. To further understand the mechanism of IFITM3 restriction, two-colour STED was used to visualize and quantify the clustering of IFITM3 in influenza-positive endosomes [112]. The study showed that IFITM3 clustering increased following influenza virus uptake, with IFITM3 coating early and recycling endosomes that contained viral NP (i.e., pre-fusion), potentially preventing virus infection at an early timepoint. A recent study demonstrated the use of novel photo-switchable probes for SMLM called Super Beacons and provided the first observation of IFITM1 nanoscale organisation at the plasma membrane [113]. The nanoscale organisation and localization of viral restriction factors during infection may then be revealed by SRM and help understand the underlying mechanism of restriction.

Uncoating is a critical step in viral replication essential for release of the viral genome into the cytoplasm of a new host cell. However, this release exposes the vRNA to cellular innate immune sensors that may initiate interferon responses and viral restriction. Influenza virus is able to suppress the RIG-I-mediated interferon response through a specific amino-acid motif in the viral polymerase proteins PB1/PA [114]. Multi-colour 3D STORM revealed a close interaction of the vRNPs with the innate immune sensors RIG-I and MAVS at the mitochondrial membrane early in infection. Therefore, nanoscale imaging suggests a model of direct molecular interactions between vRNPs and intracellular innate immune sensors during viral infection. 

Advanced microscopy, and especially quantitative microscopy, may also offer insights into the effects of antiviral drugs at the single-virus level. By studying the structures of live influenza viruses shed from cells treated with the NA inhibitor oseltamivir, quantitative microscopy and the use of non-invasive labelling showed that oseltamivir preferentially decreases the release of filamentous viruses as opposed to smaller spherical ones [22]. This suggests that the pleiomorphic diversity of influenza may help improve viral escape when challenged with oseltamivir, highlighting the potential importance of this diversity in vivo (see Figure 1b). This highlights the capabilities of advanced microscopy to relate structure to function, in the design and characterisation of anti-viral drugs.

In the context of drug development for influenza antiviral therapy, high-throughput microscopy approaches offer great promise for quantitative and large-scale analysis of compounds in the search of specific cellular and viral phenotypes. For instance, a high-throughput quantitative imaging methodology was developed to provide assays capable of probing different stages of the early influenza infection: Initial binding, endocytosis, viral membrane fusion, uncoating, nuclear import and translation [115]. The imaging approach in these assays is conventional; for example, looking at the position and intensity of anti-M1 signals to monitor uncoating. The novelty comes with the authors’ use of machine learning to automate and quantify the different visual endpoints and to optimize the timing of the read-outs. One key application of such technology is the identification of host factors as drug targets for the prevention and/or treatment of influenza.

## 8. Limitations of SRM Approaches

Despite the demonstrated applications of advanced microscopy presented here, the full potential of these techniques remains limited by shortcomings that are necessary to understand. In particular, it is important to note that, although resolution with SMLM can reach ~20 nm, an observed colocalization at such a spatial range does not directly indicate molecular interactions. SRM techniques can only suggest spatio-temporal interactions and require confirmation via biochemical or FRET assays.

Another clear limitation of SRM is its compatibility with live-cell imaging. Although technically feasible, performing live-cell imaging at molecular resolution (< 50nm) remains challenging due to both the acquisition times required (as in the case of SMLM) and phototoxicity (as in the case of both SMLM and STED). For instance, due to the technical nature of SMLM, exposure times of the order of minutes to hours are needed for optimal image reconstruction, thus limiting the observation of fast biological processes. In fact, the live-cell fPALM studies that demonstrated nanoscale cluster dynamics [44] could only follow individual molecules within clusters over short time scales and not the clusters themselves. 

Phototoxicity can also be limiting when studying light-sensitive phenomena with SRM, notably for SMLM and STED [116,117,118]. SIM, on the other hand, is more favourable to live-cell imaging due to lower laser power needed and with amenable probes and dyes that do not disrupt the normal behaviour of a sample, albeit at a resolution limited to 100–150 nm. The developments of light-sheet-based systems, capable of minimising sample exposure to illumination light, with SRM capabilities [25,119] remains an active field of research and is not yet commonly available. Light-sheet microscopy tends to provide lower resolution imaging than wide-field or confocal imaging but at faster speeds and with greater possibility for long-term imaging.

Additionally, with SRM, greater attention should be paid to the methods and types of fluorescent labelling used. When the instrument achieves resolutions of the order of the size of the label or its linker (a single IgG antibody is of the order of 10–15 nm long), then the data should be interpreted by keeping in mind that what is visualised is the marker and not the molecule of interest. The effect of linker length can have a significant effect on the quantification of structural parameters derived from SRM [120,121]. Another concern in labelling for SRM is the necessity for a labelling strategy that is sufficiently spatially dense to describe the structure of interest which cannot always be achieved due to steric hindrance. The development of short and efficient labelling for SRM is a fast-developing field of research [122,123].

Finally, the availability of SRM and other advanced microscopy systems is still a limitation preventing researchers from leveraging the power of SRM and advanced imaging. This is especially true for live imaging of viral replication which may require high levels of containment for safety reasons.

## 9. Conclusions

Our review summarizes recent applications of SRM and quantitative advanced microscopy techniques to study influenza virus replication. It highlights important techniques such as SRM for structural studies of molecular assemblies either in the virus particles [22] or in interactions with host-cell components [30], SPT via QD labelling for tracking and trafficking dynamics [64], single-molecule imaging via smFRET to study conformational dynamics of HA [66] and MuSeqFISH) to study the bundling of viral genome segments during assembly [75], all of which provided important new mechanistic insights into influenza virus replication. In addition, the application of live-cell approaches, in particular SPT of different viral components (such as vRNPs) [64] and single-virus studies [19,59], enabled by novel labelling strategies such as QD and small enzymatic tagging, have uncovered important dynamic behaviours of influenza virus infection in cells, highlighting the complexity of trafficking and assembly [28,75], and the shape-biased effect of oseltamivir on pleiomorphic viruses [22]. This diversity can only be uncovered by advanced microscopy techniques capable of high sensitivity and molecular specificity, combined with high spatial and temporal resolution.

An important underlying principle to all these microscopy methods is the combination of imaging with quantitative data analysis to determine, e.g. the relative levels and spatial distributions of envelope proteins in filamentous viral particles [39] or the spatio-temporal behaviour of endosomal vesicles and vRNA by SPT [24]. Quantitative approaches [22,24,75,115] and the study of viruses at the population level, especially those using novel machine-learning-based methods, have the potential to make a transformative change in how we analyse imaging data [70,71] in an objective and unbiased approach, especially as they become more widely available [124]. Several computational approaches have already been developed to build and map nanoscale structural features of virus morphology [21,25,125]. The strength of these approaches is that they can be combined with the microscopy techniques discussed above to extract as much information from imaging data as possible. For instance, single-molecule tracking can be performed in combination with SRM imaging of the host cell cytoskeleton or endosomal network, to precisely assign observed features to underlying cellular structures. As we have highlighted throughout this review, a number of approaches, such as the use of ExM, remain under-exploited both for the study of viral assembly and the modulation of host organelle organisation in infection.

For the first time, these new imaging technologies allow us to follow the fate of individual virions in live cells, to understand how the choice of entry pathway or assembly may be modulated depending on the cellular (e.g., activation or suppression of relevant host factors including restriction factors) and viral context (e.g., virus composition, multiple concomitant infections), and to quantify the effect of potential anti-viral drugs at the single-virus level. We expect the combination of these advanced imaging technologies with new labelling methods, and new computational approaches, including machine learning, will reveal new knowledge of influenza virus replication enabling the development of effective therapeutics. These technical advances will be important tools in understanding virus-host interactions that determine infection outcomes and pandemic potential.

## Figures and Tables

**Figure 1 viruses-13-00233-f001:**
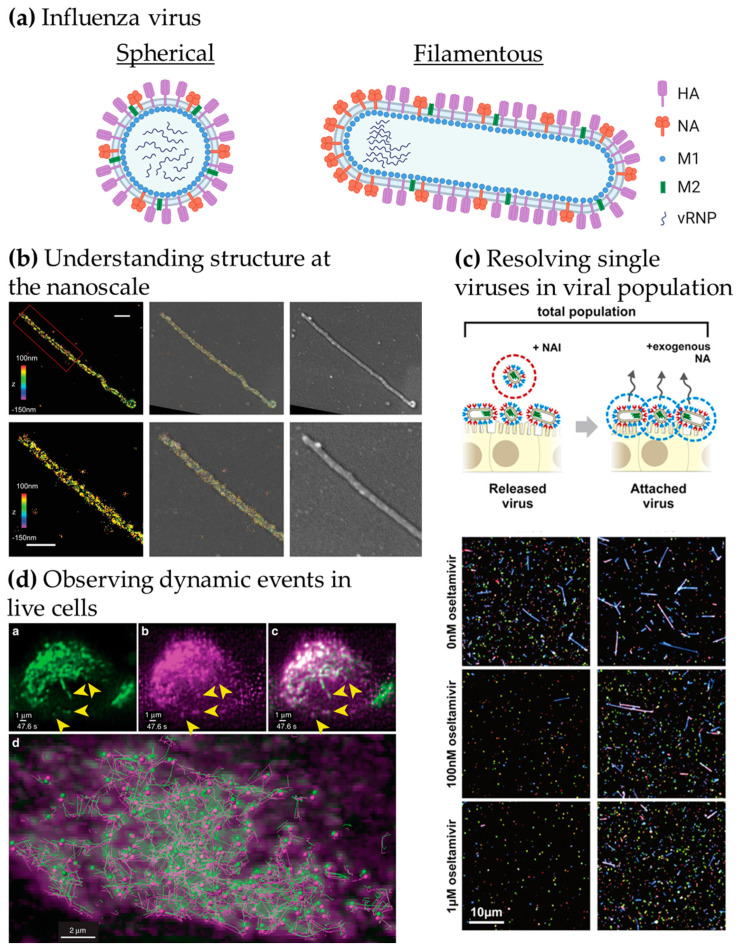
Advanced microscopy gives structural and functional insights into influenza virus replication. (**a**) Structure of influenza virions. Spherical particles diameter ~120 nm; filamentous particles length up to 20 µm [8,9]. (**b**) Understanding viral structure at the nanoscale. Correlative 3D-STORM and scanning electron microscopy (SEM) images of filamentous Udorn influenza virus immuno-labelled for HA. Left: STORM image. Right: SEM image. Middle: Overlaid image. Bottom: Magnified views of the region of interest shown on top. Scale bars: 500 nm. Adapted from [19]. (**c**) Resolving single viruses in viral populations. Influenza-infected cells were treated with oseltamivir, a neuraminidase inhibitor at different concentrations and the “Released” viruses and the viruses that remained “Attached” were imaged. Adapted from [22]. (**d**) Observing dynamic events in live cells. Live-cell light-sheet three-dimensional (3D) imaging of cells expressing GFP-Rab11A were infected with recombinant influenza expressing mRuby-PA as a marker for the vRNP. Top, left to right: GFP-Rab11A, mRuby-PA, overlay (arrows: colocalizing puncta). Bottom: colocalizing puncta of GFP-Rab11A and mRuby-PA tracks over time. Adapted from [24].

**Figure 2 viruses-13-00233-f002:**
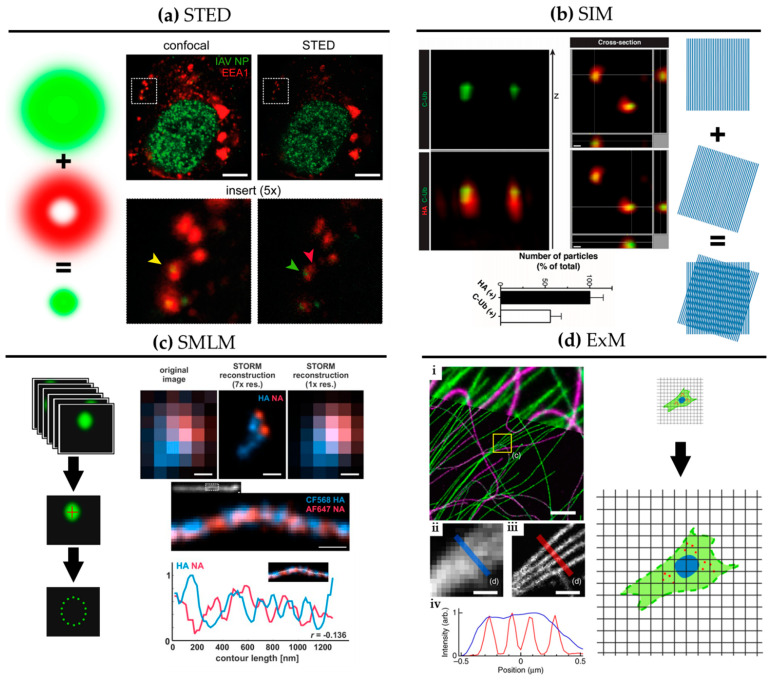
Super-resolution microscopy for the study of influenza virus replication. (**a**) STED. Confocal (left) and STED (right) images of a dendritic cell 4 hours post-infection with influenza virus vRNP (green) and the early endosomal marker EEA1 (red). Scale bar: 5 μm. Adapted from [28]. (**b**) SIM. 3D SIM z-stack and cross-section images of purified influenza viruses labelled for hemagglutinin (HA) (red) alongside immuno-labelling of the C-terminus of ubiquitin (C-Ub) (green). Bottom: quantification of HA and C-Ub colocalization. Scale bar: 100 nm. Adapted from [30]. (**c**) SMLM. Top: diffraction-limited image and *d*STORM reconstructions at ~30 nm resolution of influenza viruses labelled for NA (red) and HA (blue). Bottom: *d*STORM reconstructions of a filamentous virion labelled for HA and NA at ~30 nm resolution with corresponding intensity profile. HA and NA are shown to exclude one another. Scale bar: 200 nm. Adapted from [39]. (**d**) ExM. BS-C-1 cell labelled for tyrosinated tubulin (green) and de-tyrosinated tubulin (magenta). (i): Comparison of ExM image (bottom) with corresponding pre-expansion image (top). Scale bars: 2 µm. Insert, pre-(ii) and post-expansion (iii) with corresponding intensity profiles (iv). Scale bars: 500 nm. Adapted from [40].

**Figure 3 viruses-13-00233-f003:**
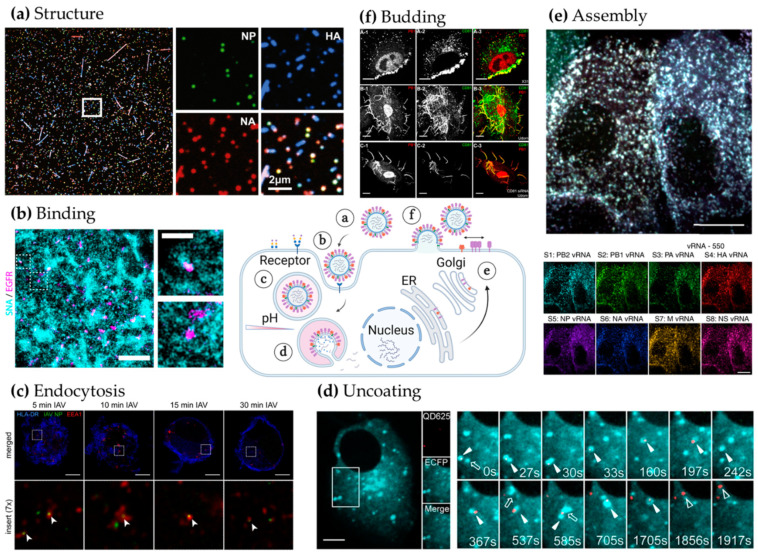
Analysis of influenza virus replication using advanced microscopy. Scheme of the influenza life cycle and corresponding microscopy images showing, anti-clockwise: (**a**) Virus particle structure and composition: Spherical and filamentous influenza virions labelled for NP (green), HA (blue) and NA (red) and immobilized on a coverslip exhibit diverse morphologies and composition. Adapted from [22]. (**b**) Binding: Left: Two-colour STORM image of A549 cells labelled for sialic acid (SNA) (blue) and anti-EGFR (magenta). Scale bar: 500 nm. Right: insert of left. Scale bar: 200 nm. Adapted from [26]. (**c**) Endocytosis: STED images of dendritic cells labelled for HLA-DR (blue), influenza virus NP (green) and EEA1 (red). Arrows point to nanoscale colocalization of EEA1 and NP signals. Scale bar: 5 μm. Adapted from [28]. (**d**) Uncoating: MDCK cells expressing Rab7-ECFP (blue) infected with influenza vRNPs labelled with a quantum dot QD625 (red). Scale bar: 5 µm. Snapshots of tracked influenza and Rab7 over time. Filled triangles: colocalization of QD and Rab7 signals, hollow triangles: released vRNPs, arrows: virus-negative endosomes. Adapted from [64]. (**e**) Assembly: bottom: each of the 8 segments of viral genomic RNA is visualised by FISH using Atto550 as label. Top: overlay. Scale bar: 10 µm. Adapted from [75]. (**f**) Budding: Top row: A549 cells were infected with influenza virus X-31 for 16 h and labelled with anti-CD81 (green) and anti-PB1 (red). CD81 is recruited to the virus budding zone in X-31 infected cells. Middle row: A549 cells infected with Udorn virus. Same staining as top. CD81 is incorporated into budding filamentous virions of Udorn-infected cells. Bottom row: same as middle row, but in CD81-KO cells. Remaining CD81 in CD81-KO cells is incorporated into budding filamentous viruses of Udorn infected cells. Scale bar: 10 µm. Adapted from [76].

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
