# Peer review of "Application of Super-Resolution and Advanced Quantitative Microscopy to the Spatio-Temporal Analysis of Influenza Virus Replication"

_viruses, 2021, doi:10.3390/v13020233_

Round 1

Reviewer 1 Report

Overview

The review describes how using super-resolution and advanced quantitative microscopy enabled key advances to the understanding of influenza A virus lifecycle. It provides examples of articles that have used these sophisticated techniques and explores how they contributed to advancing the field, but on occasion overemphasizes the conclusions reached by their usage. There are cases in which their usage was absolutely innovative and critical, as for example in observing influenza A virus entry and transport of endocytosed virions inside cells. There are other cases, where its usage has provided support to already proposed models. The distinction of the two is not clear in this manuscript. Nevertheless, the review is well written, well organized and structured, but, for the reason mentioned, could be more insightful. In addition, it does not explore the caveats of the methods and what future improvements in these methods could be made to better tackle influenza A virus lifecycle.

Major issues

My biggest concern is that, on occasion, the way authors explain the contributions of super-resolution/ quantitative approaches to the field of influenza is misleading. There are many cases in which the knowledge that the authors claim as obtained by using these approaches had been proposed before (see below).

Examples:

Page 6, line 187

“Interestingly, some of the earliest fPALM experiments looked at the association of influenza virus HA with lipid rafts [41]. This study showed that HA forms dynamic irregular clusters of varying size in lipid rafts. Due to the small size of these microdomains at the plasma membrane, it would be very challenging to observe them with conventional fluorescence imaging techniques. Further, an early form of SMLM called ‘Blink’ [42] was used to observe that HA at the plasma membrane of infected cells forms dynamic nanodomains of around 80 nm [43]”.

The way the text is written indicates that fPALM was critical to identify that HA associated with lipid rafts. This has been known for long and topic of extensive work (see this review: Simons K, Ikonen E. Functional rafts in cell membranes. Nature. 1997;387:569–572, and work since). The text should be written in a clear form explaining that HA association with lipid rafts has been shown in the late 90s, but the dynamics of this association was not known. Early fPALM experiments were able to demonstrate that it forms dynamic irregular clusters of 80nm which is interesting because that is roughly the width of a budding virion, as determined using electron microscopy-based approaches.

Page 7, line 253

“correlative STORM and EM have made it possible to observe the nanoscale distribution of HA, M1 and vRNP in a filamentous strain of influenza virus”

This sentence is also misleading. The way it is written implies that this is the first observation of distribution of proteins at the surface of virions. However, these proteins have been distinguished by electron cryo-microscopy (for example: Calder LJ, Wasilewski S, Berriman JA, Rosenthal PB. Structural organization of a filamentous influenza A virus. Proc Natl Acad Sci U S A. 2010;107(23):10685-90) without using super-resolution approaches. Nevertheless, it is clear that quantitative and super-resolution microscopy-based approaches are advantageous to look and investigate protein structure in virions because the experimental setup is more widely available, the experiment is easier and because it is possible to screen a much higher number of virions.

12 line 483

“These studies suggested an assembly of vRNPs en route to the budding site, facilitated by the association with and modulation of Rab11A-associated vesicular transport, demonstrating the profound effect of infection on molecular and vesicular transport dynamics [24]”.

This sentence is misleading. In fact, it had been proposed before that the assembly of influenza A virus genome occurred on route to the plasma membrane – (please see Eisfeld AJ, Kawakami E, Watanabe T, Neumann G, Kawaoka Y. RAB11A is essential for transport of the influenza virus genome to the plasma membrane. J Virol. 2011;85(13):6117-26).

The effect of infection on vesicular transport had also been reported before (Vale-Costa S, Alenquer M, Sousa AL, Kellen B, Ramalho J, Tranfield EM, et al. Influenza A virus ribonucleoproteins modulate host recycling by competing with Rab11 effectors. J Cell Sci. 2016;129(8):1697-710).

The quantitative methods used provided more evidence to support both models.

Please consider explaining the contribution of multicolour single-molecule fluorescent in situ hybridization more clearly:

In 2012, the lab of Peter Palese described an experimental approach based on multicolor single-molecule fluorescent in situ hybridization using FlAsH to study the composition of viral RNAs at single-virus particle resolution. This technique enables assessing the stoichiometry of the eight vRNPs inside one virion, and determine the percentage of virions with complete genomes, in a manner that is much easier than the previously employed. Previous approaches relied on using tomography of whole virions and evaluating the size of vRNPs to infer the vRNP type. Everyone will agree that this is a far more complex, less quantitative, and less accessible technique to using FlAsH and therefore it is an important contribution. As as note, to illustrate the previous comments – even being an important step forward, it would not be fair to say that FlAsH permitted the visualization of the stoichiometry of vRNPs inside individual virions as this had been accomplished before by the labs of Kawaoka and Marquet.

A second concern is that the authors do not explore the caveats of using these techniques or how they need to be complemented with other experiments to be able to answer specific questions. For example, when talking about influenza A virus assembly, imaging is not enough as co-localization in a cellular location just means that the molecules are in the same place. It does not mean that they are interacting or forming a complex.

The review would become more insightful if these topics would be addressed.

Reviewer 2 Report

Authors excellently reviewed on application of super-resolution and advanced quantitative microscopy can be used for the spatio-temporal analysis of influenza virus replication. This manuscript is worht being published in Viruses.

Author Response

We thank the reviewer for this very positive appraisal of our review.